bioengineering/biomechanics

dragonfly, flapping wing, hovering, tandem-wing interactions, phase difference

**Author for correspondence:**
Tianyu Pan
e-mail: pantianyu@buaa.edu.cn

# Tandem-wing interactions on aerodynamic performance inspired by dragonfly hovering

Liansong Peng[1], Mengzong Zheng[1], Tianyu Pan[2], Guanting Su[1] and Qiushi Li[1,2,3]

[1]School of Energy and Power Engineering, Beihang University, Beijing 100191, People's Republic of China
[2]Research Institute of Aero-Engine, Beihang University, Beijing 100083, People's Republic of China
[3]Key Laboratory of Fluid and Power Machinery, Ministry of Education, Xihua University, Chengdu 610039, People's Republic of China

TP, 0000-0002-9158-8262

Dragonflies possess two pairs of wings and the interactions between forewing (FW) and hindwing (HW) play an important role in dragonfly flight. The effects of tandem-wing (TW) interactions on the aerodynamic performance of dragonfly hovering have been investigated. Numerical simulations of single-wing hovering without interactions and TW hovering with interactions are conducted and compared. It is found that the TW interactions reduce the lift coefficient of FW and HW by 7.36% and 20.25% and also decrease the aerodynamic power and efficiency. The above effects are mainly caused by the interaction between the vortex structures of the FW and the HW, which makes the pressure of the wing surface and the flow field near the wings change. During the observations of dragonfly flight, it is found that the phase difference ($\gamma$) is not fixed. To explore the influence of phase difference on aerodynamic performance, TW hovering with different phase differences is studied. The results show that at $\gamma = 22.5°$, dragonflies produce the maximum lift which is more than 20% of the body weight with high efficiency; at $\gamma = 180°$, dragonflies generate the same lift as the body weight.

## 1. Introduction

Most flying insects possess only one pair of wings (e.g. Diptera and Strepsiptera), or have their forewings (FWs) and hindwings (HWs) in contact (e.g. Hymenoptera and Lepidoptera) which cannot move independently [1]. Dragonflies are among the few

insects that can control the kinematics of their four wings independently with direct musculature at each wing base to deal with complex flight conditions. They can hover [1], cruise up to 54 km h$^{-1}$ [2], turn 90°–180° in two or three wing beats [3,4], fly sideways, fly backwards [5] and glide [6]. They can produce total aerodynamic force equal to 4.3 times of the body weight [7] and are the only found insect capable of transoceanic immigration [8]. Interactions between FW and HW have an important effect on aerodynamic performance of dragonflies. The phase difference is an important kinematic parameter for the interaction which is defined as the phase angle by which the HW leads the FW. According to the observations of dragonfly flight [9–17] and the studies of phase difference [18–20], it was concluded that dragonflies could make various kinds of interactions by adjusting phase difference in different flight conditions, and the change of phase difference had an impact on aerodynamic performance of tandem-wing (TW) flapping. Adjusting phase difference in a TW flapping might be an outstanding method to control flight performance.

Hovering is a common flight condition for dragonflies, which is defined as flight with zero net velocity relative to the air. To maintain the hovering flight, dragonflies keep their bodies relatively still and the kinematics of their wings basically periodic to provide a constant upward force equal to the body weight. Therefore, researchers made numerical and experimental efforts to investigate the aerodynamic mechanism of TW hovering like a dragonfly [21–24]. Earlier studies mainly focused on the aerodynamic force generated by TWs and concluded that the TW interactions caused a decrease in lift. The TW interactions reduced the vertical forces on the FWs and HWs by 14% and 16%, respectively [22]. However, there is a lack of systematic research on the effects of TW interactions on the hovering efficiency. To our knowledge, Usherwood & Lehmann [25] used a mechanical model to investigate the hovering efficiency with TW flapping. Their results showed that with appropriate phase difference ($\gamma = 45°$), aerodynamic power could be decreased and hovering efficiency could be increased compared to the flapping with a single pair of wings. Lian [26] used a numerical method to study two-dimensional TW flapping. The simulation results showed that when $\gamma = 0°$ and 90°, the hover efficiency was less than that of single-winged flapping, and when $\gamma = 180°$, the hover efficiency was greater than that of single-wing flapping. Nagai [20] investigated the effect of phase difference in TW hovering experimentally. The results showed that the vertical force efficiency for the total wings in tandem is smaller than that in isolation, except for $\gamma = 0°$ and 45°; the maximum hovering efficiency for the total wings in tandem at $\gamma = 0°$ is 4.3% larger than that in isolation. Therefore, it is necessary to reach a uniform conclusion by systematically studying the influence of TW interaction on hovering efficiency and the influence of phase difference on the aerodynamic performance during hovering. As such, the two objectives of the current study are to investigate the effects of TW interactions on hovering efficiency and to investigate the effect of phase difference on TW hovering.

In this paper, a numerical simulation method is used to simulate the TW hovering. By comparing the aerodynamic parameters of single-wing hovering and TW hovering with the kinematic law of dragonfly hovering at $\gamma = 180°$, the aerodynamic effects of TW interactions on hovering are obtained. Combined with the analyses of flow fields, the mechanism of how the TW interactions affect hover aerodynamic performance is investigated. By comparing the efficiency and power under different phase differences from 0° to 180°, the influence of phase difference on dragonfly hovering is studied.

# 2. Material and methods

## 2.1. Wing geometry and kinematics

The orientation of the wing is determined by three Euler angles with respect to the stroke plane. The stroke plane based on the wing tip line and the root of the wing is shown in figure 1a. OXYZ is an inertial frame with the X and Y axes in the horizontal plane and point O is the centre of mass. The definition of the three Euler angles is shown in figure 1b: the positional angle $\phi$, the rotational angle $\theta$, and the stroke deviation angle $\psi$; where $\phi$ is defined as the angle between the Z-axis and the projection onto the stroke plane of the line joining the wing base and wing tip, $\theta$ is defined as the angle between the local wing chord and line $l$ ($l$ is perpendicular to the wing span and parallel to the stroke plane), and $\psi$ is defined as the angle between the line joining the wing base and the wing tip and its projection onto the stroke plane. The angle of attack $\alpha$ has the following relationship with $\theta$: in the downstroke, $\alpha = \theta$; in the upstroke, $\alpha = 180° - \theta$.

Norberg [28] first filmed dragonfly hovering in the field by high-speed camera. Through analysing the photos, he found that the phase difference of the dragonfly hovering was 180°. This result provided

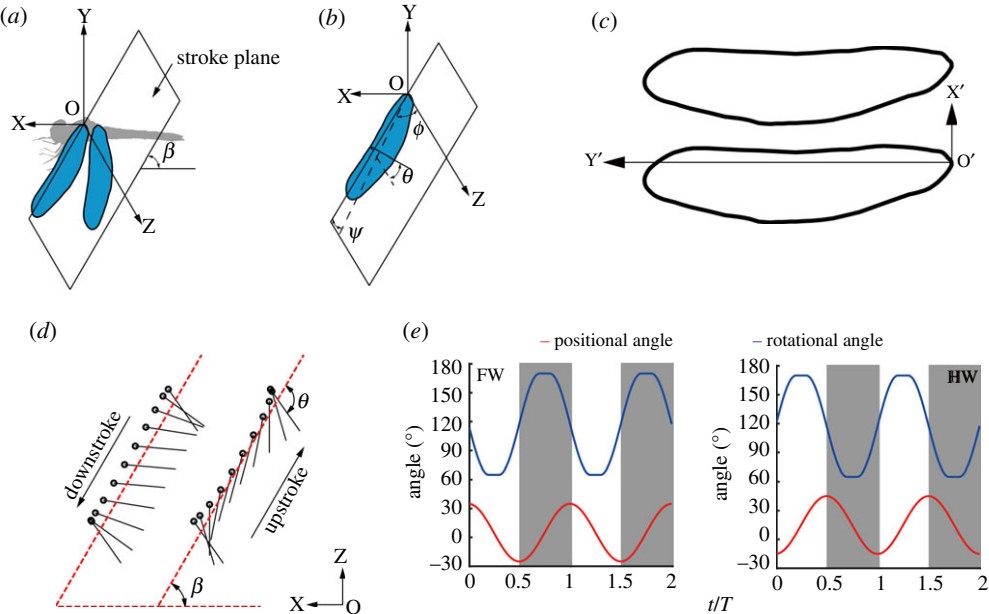

**Figure 1.** Definition of kinematics and geometry of the FW and HW used in the simulation. (*a*) Stroke plane and stroke plane angle $\beta$ for describing the flapping trajectory. (*b*) Three Euler angles of the kinematics of the wing with respect to the stroke plane: the positional angle $\phi$, the rotational angel $\theta$, and the stroke deviation angle $\psi$. (*c*) The geometry of wings referred to the measurement of Norberg [27]. (*d*) The two-dimensional diagram of wing motion. (*e*) The instantaneous Euler angles for FW and HW for two periods.

a basis for the subsequent experimental and numerical work, while with further observation of dragonflies, the researchers found that the phase difference in their hovering was not constant. In most cases, dragonflies hovered with $\gamma = 180°$; while in a few cases, smaller phase differences between 60° and 90° were applied [9,10]. The wing geometry and kinematics of dragonflies in this study are mainly based on data from Norberg's measurement [27,28]. By analysing the high-speed photos of hovering dragonflies (*Aeshna juncea*), he found that during hovering the body was held almost horizontal, the stroke frequency $f$ was about 36 Hz, the stroke plane angle was 60° relative to the horizontal plane, the positional angle of FW was 60° from 35° (above the horizontal plane) to −25° (below the horizontal plane), and the positional angle of HW was 60° from 45° to −15°. The mean body weight was 754 mg, the mean length of FW and HW were 4.74 cm and 4.60 cm, mean chord length of FW and HW were 0.81 cm and 1.10 cm. The same wing geometry is adopted for the FW and the HW in this paper. The planforms of FW and HW in this simulation shown in figure 1*c* are similar to those of the real wings, where the thickness of wing is set to $0.01c$ (the mean chord length). O′X′Y′Z′ is a frame fixed on the wing, with the X′ axis along the wing chord and Y′ axis along the wing span and point O′ is the wing root.

It is widely accepted that the positional angle can be well represented by a first harmonic function [10–12]. By interpolating Norberg's data, the positional angle is given by

$$\phi(t) = \frac{\phi^+ - \phi^-}{2} + \frac{\phi^+ + \phi^-}{2}\cos(2\pi T + \lambda), \qquad (2.1)$$

where $\phi^+$ and $\phi^-$ are extreme positional angles. The detailed kinematics of the positional angle $\phi(t)$ is as follows:

$$\phi(t) = 5° + 30°\cos(2\pi T). \qquad (2.2)$$

According to the observations of dragonflies [11–13] and other insects [27], the rotational angle remains almost a constant value in the middle portion of a half-stroke and changes as sinusoidal function around the stroke reversal. Time is expressed as a non-dimensional parameter, $T$. $T = 0$ represents the start of the downstroke and $T = 1$ represents the end of the upstroke. Therefore, the rule

of rotational angle is expressed as follows by piecewise function:

$$
\theta(t) = \begin{cases}
\dfrac{\theta_u + \theta_d}{2} + \dfrac{\theta_u - \theta_d}{2}\sin\left[\pi\left(\dfrac{T}{\Delta T}\right)\right] & \left(-\dfrac{\Delta T}{2}, \dfrac{\Delta T}{2}\right) \\[2ex]
\theta_d & \left(\dfrac{\Delta T}{2}, \dfrac{1 - \Delta T}{2}\right) \\[2ex]
\dfrac{\theta_u + \theta_d}{2} + \dfrac{\theta_u - \theta_d}{2}\sin\left[\pi\left(\dfrac{1/2 - T}{\Delta T}\right)\right] & \left(\dfrac{1 - \Delta T}{2}, \dfrac{1 + \Delta T}{2}\right) \\[2ex]
\theta_u & \left(\dfrac{1 + \Delta T}{2}, 1 - \dfrac{\Delta T}{2}\right)
\end{cases}
\tag{2.3}
$$

where $\Delta T$ is the non-dimensional time interval over which the rotation lasts. Set $\Delta T = 0.4$, upstroke rotational angle $\theta_u = 170°$, downstroke rotational angle $\theta_d = 65°$, phase difference $\gamma = 180°$ according to the data in [12,13,28]. When $-0.2 < T < 0.2$, the wing pronates from $\theta_u$ to $\theta_d$; when $0.3 < T < 0.7$, the wing supinates from $\theta_d$ to $\theta_u$; when $0.2 < T < 0.3$, the rotational angle keeps $\theta_d$ during downstroke; when $0.7 < T < 0.8$, the rotational angle keeps $\theta_u$ during upstroke.

A diagram of the motion of FW is shown in figure 1$d$. In this paper, the positional angle and rotational angle adopted by the FWs and the HWs are consistent. The time courses of kinematics of FW and HW for the hovering dragonflies are shown in figure 1$e$. The upstrokes of FW are shaded in grey.

The radius of the second moment is denoted by $r_2 = \int_S r^2 \cdot dS/S$, where $r$ is radial distance from wing root and $S$ is the area of the wing. The reference velocity $U_{ref}$ is the average positional velocity at $r_2$. On the basis of the above data, the Reynolds number of dragonfly flapping is 1196, which is defined as $\mathrm{Re} = (U_{ref}\,c)/v$, where $v$ is the kinematic viscosity coefficient of air.

## 2.2. Numerical simulation method

Computational fluid dynamics simulations are conducted using the commercial solver Xflow 2019 with the lattice Boltzmann method (LBM). Unlike conventional numerical schemes based on discretization of macroscopic Navier–Stokes equations, LBM is based on microscopic models. LBM works on a spatial discretization named lattice, consisting of a Cartesian distribution of discrete points with a discrete set of velocity directions $e_i$ ($i = 1, \ldots, b$), and has successfully simulated a range of flow conditions, including porous media, human blood flow, vortex shedding, multiphase flows, droplet dynamics and turbulent flows [29–36]. The Boltzmann transport equations in the continuum space can be written as follows:

$$
\frac{\partial f_i(R,t)}{\partial t} + e_i \cdot \nabla f_i(R,t) = \Omega_i, \quad i = 1, \ldots, b,
\tag{2.4}
$$

where $f_i$ ($R$, $t$) is the distribution function for particles with velocity $e_i$ at position $R$ and time $t$, $\Omega_i$ is the collision operator that computes a post-collision state conserving mass and linear momentum. Equation (2.4) can be discretized as

$$
f_i(R + e_i) = f_i(R,t) + \Omega_i(f_1,, \ldots, f_b), i = 1, \ldots, b.
\tag{2.5}
$$

The macroscopic density and linear momentum can be computed as

$$
\rho = \sum_{i=1}^{b} f_i \quad \text{and} \quad \rho v = \sum_{i=1}^{b} f_i e_i.
\tag{2.6}
$$

Since the Cartesian distribution of lattice is used in LBM, the meshing process is greatly simplified, which means LBM has advantages over conventional numerical methods in solving moving boundary conditions with complex surfaces. The three-dimensional, single-phased flow model with external analysis is used to simulate the hovering. The particle-based, fully Lagrangian approach is used to efficiently solve the incompressible flow. The unified non-equilibrium wall model is used to accurately model the boundary layer. The adaptive refinement is used to refine in regions close to the boundary surface and the wake generated. The behaviour of the wings is set as enforced so that the kinematic rules can be accurately achieved by defining the Euler angles.

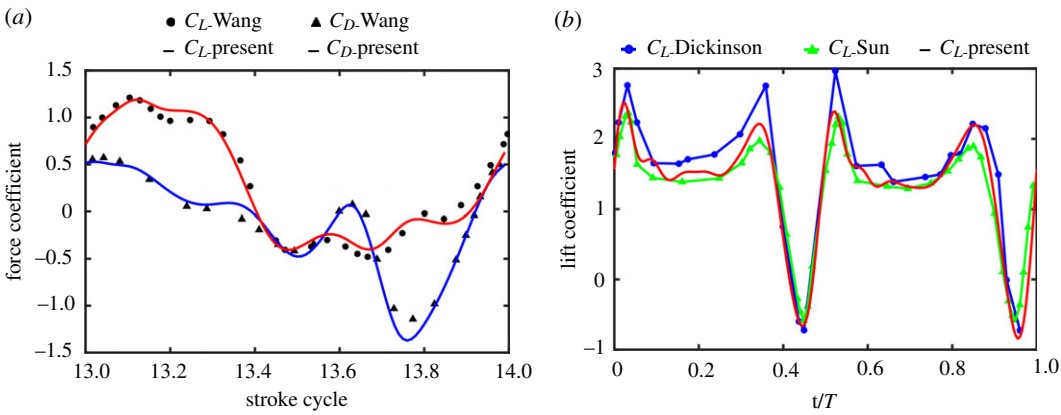

**Figure 2.** Validation of numerical method. (*a*) Comparison with a two-dimensional dragonfly hovering data by Wang [37]. (*b*) Comparison with three-dimensional experimental and numerical data from Dickinson [38] and Sun [39], respectively.

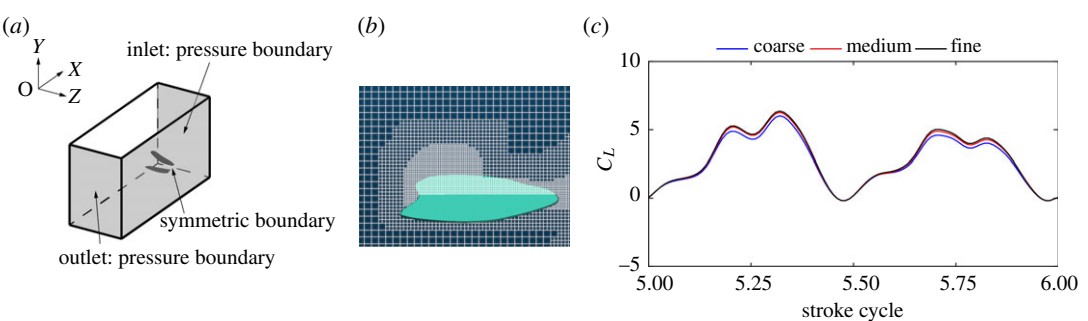

**Figure 3.** (*a*) Computational domain and boundary conditions. (*b*) The distribution of dynamically refined lattices at $r_2$ section. (*c*) Lift coefficient computed with coarse, medium and fine grids.

## 2.3. Validation

To validate the present numerical method, two typical flows of flapping wings are examined. Figure 2*a* shows the time courses of horizontal and vertical force coefficient during the fourteenth stroke for dragonfly hovering flight, which has been carefully studied by Wang [37]. Figure 2*b* shows the time courses of vertical force coefficient during a whole cycle for fruitfly hovering, which has been researched by Dickinson [38] and Sun [39] through model experiment and numerical simulation, respectively. It can be seen that the present results agree well with data obtained by this research. Therefore, the numerical simulation method used in this paper can accurately simulate the flow of flapping insects.

Figure 3*a* and *b* show the boundary conditions and the distribution of lattices at $r_2$ section, respectively. To accurately capture the flow near the surface of the wings, dynamically refined lattices are applied near wings and vortexes. The surfaces of the wings are modelled as non-slip walls, the inlet and outlet boundary conditions are gauge pressure outlet. In order to save computational resources, only the wings on the left side are simulated in this paper. By setting lateral boundaries as symmetric, the influence of the right wings on the left wings is considered.

Before further simulation of dragonfly hovering, a lattice refinement study is conducted to determine an optimal size mesh that can provide a sufficiently accurate solution of the three-dimensional flow near the wings with the kinematics described in §2.1. The unsteady flow near the flapping wings is computed with coarse, medium and fine resolutions containing 2.05, 3.84 and 8.42 million grid points, respectively. Figure 3*c* shows the total lift under the same hovering condition, which have been obtained on coarse, medium and fine grids. As follows from this comparison, the medium size mesh provides a lift that is nearly the same as that computed with the fine mesh. Therefore, the medium size mesh with a target resolved scale of 0.001*c* is used for the cases. The dimensions of the computational domain are set to $30c \times 20c \times 10c$ and the number of time steps per cycle is 600. When further refinements on domain and time step are applied, the variation of mean total lift is less than 0.5%.

## 2.4. Evaluation of the aerodynamic forces, power and efficiency

The lift $L$ and the thrust $T$ are defined as the period-averaged vertical and horizontal component of the aerodynamic force on the wing, respectively. To distinguish the FW from the HW, forces are donated as $L_F$, $T_F$ for FW and $L_H$, $T_H$ for HW.

The aerodynamic force of dragonflies is provided by the FWs and HWs, so the total lift $L_{total}$ and the total thrust $T_{total}$ are denoted as

$$L_{total} = L_F + L_H \quad \text{and} \quad T_{total} = T_F + T_H. \tag{2.7}$$

The aerodynamic power $P_a$ along each motion axis is obtained by multiplying the moments about the axis of $X'$, $Y'$ and $Z'$ with relevant angular velocities. The results are summed to obtain total aerodynamic power as

$$P_a = M_a \cdot \Omega, \tag{2.8}$$

where $M_a$ is the aerodynamic torque and $\Omega$ is the angular velocity.

The lift, thrust and power coefficients are calculated for each wing individually as

$$C_L = \frac{L}{0.5\rho S U_{ref}^2} \quad \text{and} \quad C_T = \frac{T}{0.5\rho S U_{ref}^2} \quad \text{and} \quad C_P = \frac{P_a}{0.5\rho S U_{ref}^3} \tag{2.9}$$

where $C_L$, $C_T$ and $C_P$ are the lift, thrust and power force coefficients, $\rho$ is the fluid density.

The ratio of lift coefficient to power coefficient is taken as the measure of hovering efficiency, similar to the practice of [26,40,41]:

$$\eta = \frac{C_L}{C_P}. \tag{2.10}$$

# 3. Results

## 3.1. Total lift and thrust in dragonfly hovering

The most common phase difference in the hovering dragonfly is 180°. Therefore, in this section, 180-degree TW hovering is simulated. The calculation of TW hovering is started when the two wings flap from rest in the air and is ended when periodicity in aerodynamic forces and flow structure is approximately achieved. As time courses of total lift coefficient and total thrust coefficient shown in figure 4*a*, periodicity is achieved approximately 5–6 cycles after the calculation is started. Therefore, the simulation results of the sixth cycle are selected for data analysis. The time-average total thrust coefficient $C_{T,total}$ is 0.06, which is approximately zero. The time-average total lift coefficient $C_{L,total}$ is 1.41, which can provide lift (764 mN) that is approximately equal to the body weight (756 mg). Therefore, the hover state can be achieved with the kinematics in §2.1.

Time courses of $C_L$ and $C_T$ of tandem forewing (TF), tandem hindwing (TH) and the TW are shown in figure 4*b* and *c*. As dragonflies hover along an inclined stroke plane and the downstroke has a small rotational angle, the aerodynamic force generated during the downstroke is oriented upwards and makes the main contribution to the lift. There are two lift peaks in one cycle, which are generated by the downstroke of TF during $0.1 < T < 0.4$ and by the downstroke of TH during $0.6 < T < 0.9$, respectively. The lift coefficient peaks produced by TF and TH are 6.87 and 5.60, respectively. The TF produces 55% of the total lift and the TH produces 45% of the total lift. Thrust is mainly generated around the beginning and end of the upstroke. Different from the horizontal stroke plane applied by single-pair-wing insects, dragonflies adopt a unique inclined stroke plane in hovering, and the wing surface is almost parallel to the horizontal plane in the downstroke. Therefore, lift is mainly generated in the downstroke during which only a little thrust is generated. In the upstroke, the angle between the wing surface and the horizontal plane is relatively large, so the thrust is mainly produced in the upstroke. The direction of the thrust generated is consistent with the rotational angle which changes sinusoidally during the beginning and end of the stroke. Therefore, owing to the change of force direction, most of the positive and negative thrust are offset, resulting in little mean thrust in hovering.

## 3.2. Aerodynamic parameters of tandem-wing hovering and single-wing hovering

To investigate the effect of TW interactions in hovering, numerical simulations of single-forewing hovering (SF) and single-hindwing hovering (SH) with the same kinematics as TF and TH are

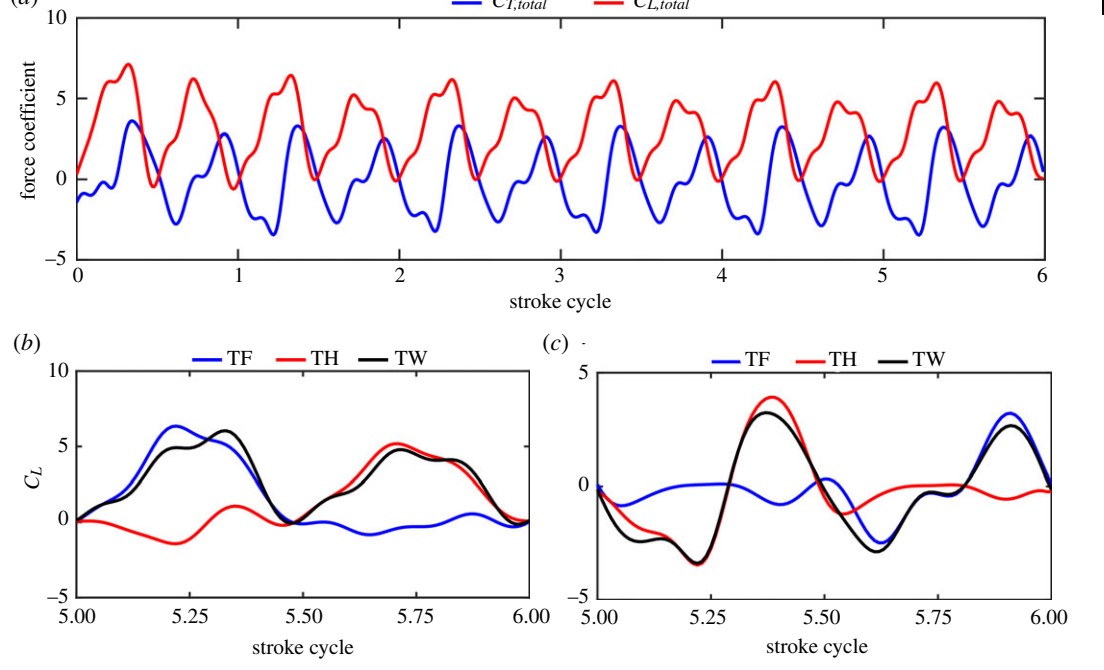

**Figure 4.** Time courses for TW hovering. (*a*) Total lift and drag for six periods. (*b*) $C_L$ of TF, TH and TW. (*c*) $C_T$ of TF, TH and TW.

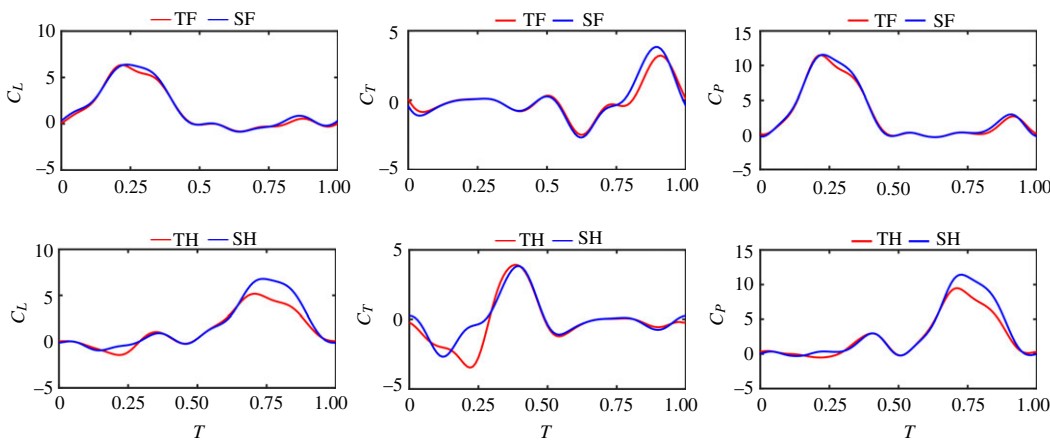

**Figure 5.** Time courses of $C_L$, $C_T$ and $C_P$ of TF/SF and TH/SH.

conducted. In this section, by comparing the aerodynamic performance of SF/TF and SH/TH, the effect of TW interactions is obtained.

Figure 5 compares the time courses of lift coefficient, thrust coefficient and power coefficient of the SF/TF and SH/TH. As $C_{T,total}$ (0.06) can be ignored compared with $C_{L,total}$ (1.41) and more than 93% of the power is used to generate lift, it is concluded that thrust has little influence on power and efficiency. Therefore, this paper mainly discusses the influence of TW interactions on lift, power and efficiency.

As shown in figure 5, the $C_L$ and $C_P$ of SF/TF and SH/TH are almost the same for most of the time in a cycle. However, when $0.2 < T < 0.4$, the $C_L$ and $C_P$ of SF is a little higher than that of TF during the downstroke of the FW; when $0.7 < T < 0.9$, the $C_L$ and $C_P$ of SH is significantly higher than that of TH during the downstroke of the HW. Therefore, it is considered that the TW interactions are obvious during $0.2 < T < 0.4$ and $0.7 < T < 0.9$ as the $C_L$ and $C_P$ of ST/TF and SH/TH are quite different and the distance between TF and TH is relatively close during the period.

The hovering efficiency $\eta$ is calculated by equation (2.10). The average value of $C_L$, $C_P$ and $\eta$ are summarized in table 1. As can be seen from the data, for both FW and HW, TW interactions reduce

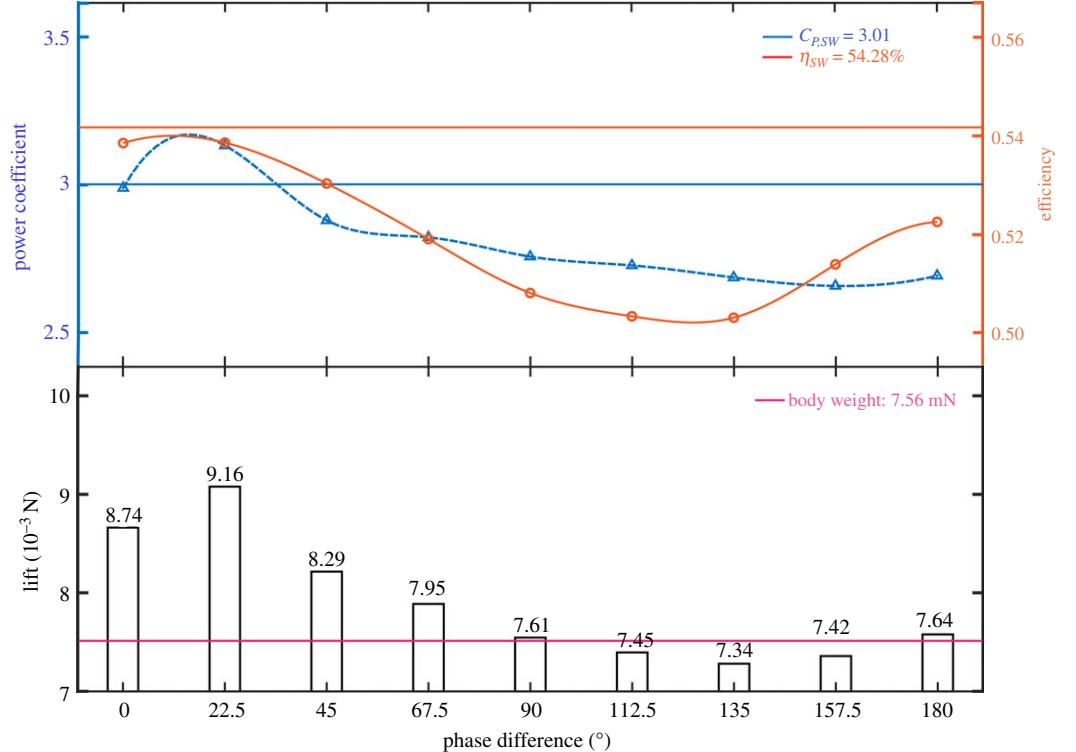

**Figure 6.** Power coefficient, efficiency and lift of TW hovering over phase difference from 0° to 180°.

**Table 1.** The aerodynamic parameters of TW hovering and single-wing hovering.

|        | SW    | TF              | TH              | TW              |
|--------|-------|-----------------|-----------------|-----------------|
| $C_L$  | 1.63  | 1.51 (↓7.36%)   | 1.30 (↓20.25%)  | 1.41 (↓13.50%)  |
| $C_P$  | 3.01  | 2.90 (↓3.65%)   | 2.49 (↓17.28%)  | 2.70 (↓10.30%)  |
| $\eta$ | 54.28 | 52.11 (↓4.00%)  | 52.44 (↓3.39%)  | 52.26 (↓3.72%)  |

the $C_L$, $C_P$ and $\eta$. The HW is more affected by the effect of TW interactions. For FW, the interactions reduce $C_L$ by 7.36% of that of SW; for HW, the reduction is 20.25%. The reduction of $C_P$ for TF and TH are 3.65% and 17.28%, respectively, compared with that of SW. The interactions decrease the efficiency of FW and HW by 4.00% and 3.39%, respectively.

## 3.3. Aerodynamic parameters of hovering with different phase differences

In most observations of dragonflies, the phase difference during hovering is 180°. However, in a few observations, the phase difference decreases to 60°–90°. To investigate the effect of phase difference on the aerodynamic performance of dragonfly hovering, this study simulates the hover with phase difference from 0° to 180°. The kinematic parameters remain unchanged except for the phase difference.

As the phase difference changes, it has little influence on thrust as $C_T$ maintains a small value, varying from 0.06 to 0.15. Therefore, the influence of phase difference on lift, power coefficient and efficiency is mainly investigated as shown in figure 6. With the increase of phase difference, the $C_P$ first increases, achieving the maximum value at $\gamma = 22.5°$, and then decreases. After the phase difference achieves greater than 67.5°, the $C_P$ remains a small value and the variation is less than 3.5%. When $\gamma = 22.5°$, the $C_P$ of TW flapping is greater than that of single-wing flapping.

The variation trend of the efficiency is similar to that of the $C_P$. When the phase difference is small, the efficiency increases with the increase of phase difference and achieves the maximum value near $\gamma = 22.5°$. However, different from the power, the efficiency increases again after achieving the minimum near $\gamma = 135°$ and reaches a local maximum at $\gamma = 180°$. When $\gamma = 0°$ to 180°, the efficiency of TW flapping is less than that of single-wing flapping

Lift achieves the maximum value at $\gamma = 22.5°$ which is greater than 20% of the body weight. Then, the lift decreases with the increase of the phase difference and is less than the body weight within the range of 112.5°–157.5°.

It can be seen that the phase difference of dragonfly hovering has two local optimal ranges: within the range of 0° to 40°, it can provide the maximum lift force which is greater than 20% of the body weight with high efficiency; at $\gamma = 180°$, the lift supporting the body weight can be provided with the minimum $C_P$. This explains why the researchers found that there were two ranges of phase differences in dragonfly hovering during observation. When dragonflies are in normal hover which needs to provide lift equal to the body weight, they apply $\gamma = 180°$ in TW flapping with low power and high efficiency to reduce energy consumption. When dragonflies need to provide more lift than their own body weight in some special hover situations with payload, such as drenched in dew or carrying prey, they reduce the phase difference to about 22.5° to provide greater lift with high efficiency.

It should be pointed out that the calculated local optimal value of phase difference (22.5°) differs from the value in the actual observation of dragonflies (60°–90°), which may be caused by the differences in the morphology and kinematics of wings.

# 4. Flow field analyses and discussion

## 4.1. The effect of TW interactions on hovering

As shown in figure 5, during the downstroke ($T = 0 \sim 0.5$ for SF/TF, $T = 0.5–1$ for SH/TH), the wing produces most of the lift and the power, while the thrust is close to zero. During the upstroke ($T = 0.5 \sim 1$ for SF/TF, $T = 0–0.5$ for SH/TH), the wing produces most of the thrust, while the lift and power are close to zero. During the middle portion of FW downstroke ($0.1 < T < 0.4$), the $C_L$ and $C_P$ of SF increase by 3.71% and 3.49% compared with that of TF, respectively; while for the rest of the cycle, SF and TF have almost the same lift and power. During the middle portion of HW downstroke ($0.6 < T < 0.9$), the $C_L$ and $C_P$ of SH are 20.85% and 22.80% higher than that of TH; while for the rest of the cycle, SH and TH have almost the same lift and power. Therefore, it is considered that the distance between TF and TH is relatively close and the TW interactions are obvious during $0.1 < T < 0.4$ and $0.6 < T < 0.9$. The flow fields at $T = 0.2$, 0.3, 0.7 and 0.8 are compared and analysed to obtain the reasons for the decrease in lift, power and efficiency.

In this section, the flow fields of the whole flapping period are analysed, and the information of the flow field at the critical time point is extracted to explain the influence of the interference of the wings on the aerodynamic performance. Through the analyses of vorticity contours of TW, SF and SH at $r_2$ section in figure 7, the strength of the trailing-edge vortex (TEV) generated by TF decreases by 17.1% in the middle part of the FW downbeat ($T = 0.3$) under the influence of TH. In the middle part of the downbeat of TH ($T = 0.8$), the strength of leading-edge vortex (LEV) generated by TH decreases by 15.3% under the influence of TF. At other times, the interactions of TWs have little influence on vorticity. Therefore, the information of flow fields at two time points with obvious interference, $T = 0.3$ and $T = 0.8$, are extracted for detailed analysis.

Figure 8a shows the flow streamlines of TW, SF and SH at $r_2$ section at $T = 0.3$. By comparing the flow streamlines of TW and SF, it can be seen that the strong TEV of TF is affected by the movement of TH in TW flapping, and part of the TEV is absorbed on the lower surface of TH, reducing the strength of the TEV. According to the streamline comparison between TW and SH, part of the wake of TF affects the air flow on the upper surface of TH, which causes the air flow to break away from the upper surface of TH and form a vortex. The disturbance produced by the wake of TF disrupts the air flow, causing energy dissipation and reducing the hovering efficiency.

Figure 8b shows the three-dimensional vortex structures of TW, SF and SH at $T = 0.3$. The Q criterion is used to depict three-dimensional vortex structures. Based on the definition in [42], Q is normalized by $Q^* = Q/(V_{\mathrm{max},2}/c)^2$ and $Q^*$ below a certain negative threshold is indicative of the vortex-dominated region. Therefore, $Q^* = -0.2$ is selected as the negative threshold. The vortex ring formed by the LEV, wing tip vortex (WTV) and TEV can generate intense downwash airflow and is the main aerodynamic structure to generate lift in the downstroke. According to the comparison of vorticity structures of TW and SF, it can be seen that under the influence of TH movement, the TEV of TF attaches to TH, which reduces the lift force of TF contributed by the TEV. It can be seen from the vorticity structures of TW and SH that part of the TEV of TF is broken and flows into the upper surface of TH, which causes flow disturbances.

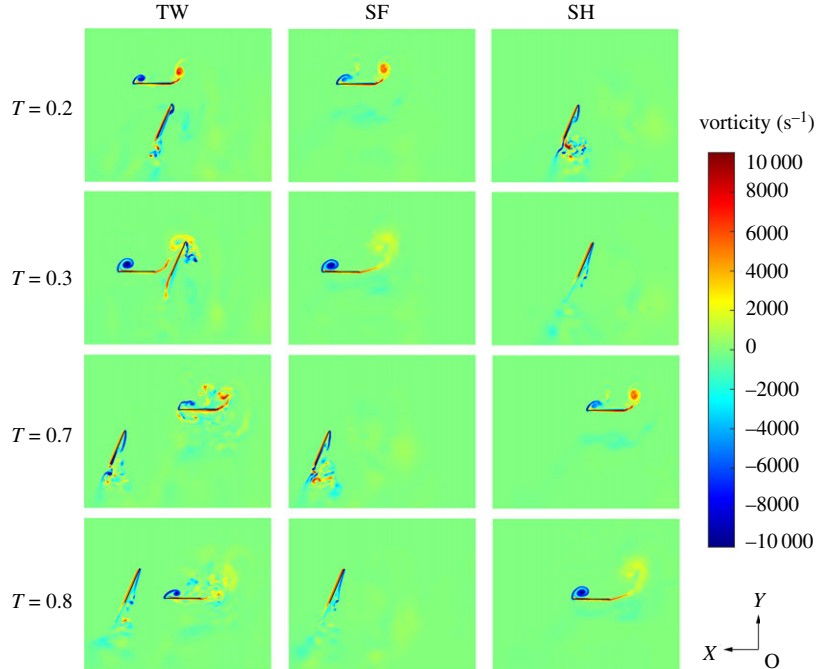

**Figure 7.** Vorticity contours of TW, SF and SH at $r_2$ section at $T = 0.2$, 0.3, 0.7 and 0.8.

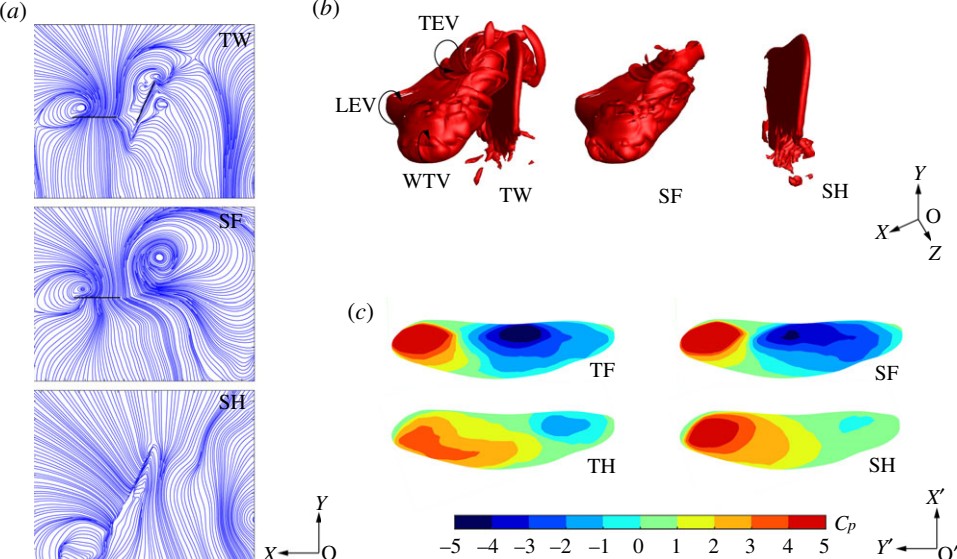

**Figure 8.** Flow fields of TW, SF and SH at $T = 0.3$ (*a*) Streamlines at $r_2$ section. (*b*) Vortex structures. (*c*) $C_P$ of upper surfaces.

According to the flow field analyses of figure 8*a,b*, the influence of TW interactions during the downstroke of TF is as follows: (i) The TEV of TF attaches to the lower surface of TH, reducing the lift of TF; (ii) The TEV of TF flows into the upper surface of TH, causing disturbance to the airflow on the upper surface of TH which reduces the efficiency. By analysing the pressure coefficient distribution on the upper surface of TF/SF and TH/SH at $T = 0.3$ in figure 8*c*, the influence of TW interactions during the downstroke of the FW can also be obtained. The low-pressure area generated by the vortex ring on the upper surface of TF is smaller than SF, indicating the strength of the vortex ring is weakened and the lift is reduced. The pressure distribution on the upper surface of TH is more chaotic than that of SH, indicating that the wake of TF interferes with the airflow on the upper surface of TH.

Figure 9*a* shows the flow streamlines of TW, SF and SH at $r_2$ section at $T = 0.8$. By comparing the flow streamlines of TW and SF, it can be seen that the LEV generated on the upper surface of TF during the

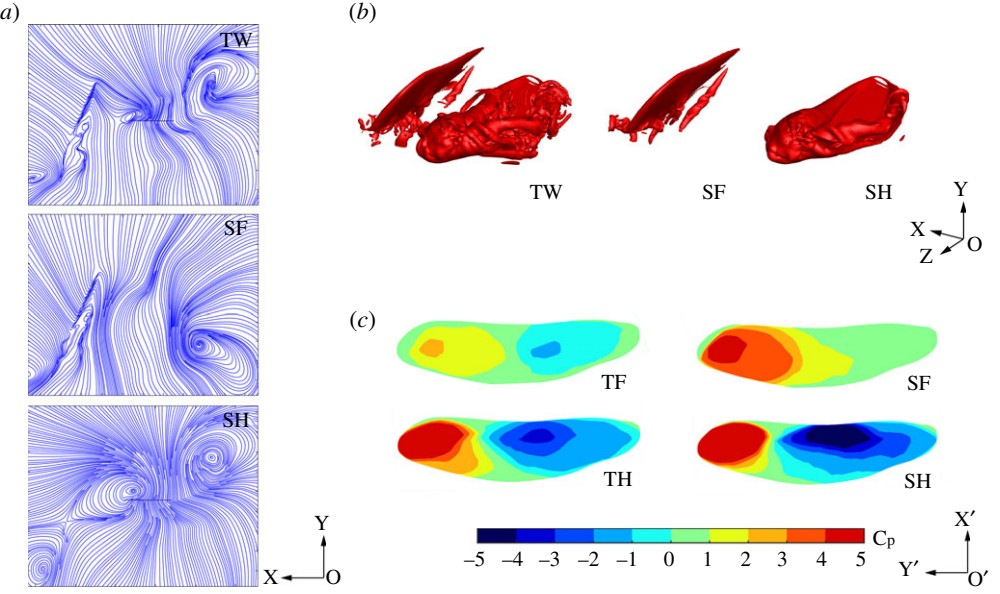

**Figure 9.** Flow fields of TW, SF and SH at $T = 0.8$ (a) Streamlines at $r_2$ section. (b) Vortex structures. (c) $C_P$ of upper surfaces.

upstroke is affected by the LEV of TH. Part of the LEV of TF is absorbed into the incoming flow of TH, which reduces the strength of the LEV. It can be seen from the streamline comparison of TW and SH that part of the LEV of TF flows into TH, which reduces the attack angle of the incoming flow of TH and weakens the strength of the LEV of TH.

Figure 9b is the three-dimensional vortex structures of TW, SF and SH at $T = 0.8$, as depicted by isosurfaces of $Q^* = -0.2$. According to the vortex structures of TW and SF, it can be seen that under the influence of TH movement, the LEV of TF is attached to TH, which reduces lift force of TF contributed by LEV. According to the vortex structures of TW and SH, it can be seen that part of the LEV of TF flows into TH, which makes the LEV of TH chaotic and weakens its strength and stability.

According to the analyses of figure 9a,b, the influence of TW interactions during the upstroke of TF is as follows: part of the LEV of TF flows into the incoming flow of the TH, reducing the attack angle of the incoming flow of TH and weakening the contribution of the LEV to lift. By analysing the pressure distribution on the upper surface of TF/SF and TH/SH in figure 9c, the influence of TW interactions can also be obtained. During the upstroke of TF, the LEV flows into the TH, making the low-pressure region of the upper surface of TF less than that of SF. The low-pressure area on the upper surface of TH is less than that of SH, indicating that the wake of TF interferes with the airflow on the upper surface of TH and weakens the contribution of the LEV of TH to the lift force.

## 4.2. The effect of phase difference on hovering

As can be seen from the summary of the lift, power coefficient and efficiency of TW hovering under different phase differences in §3.3, there are two local optimal phase differences: a small phase difference (around $\gamma = 22.5°$) which can provide high lift force efficiently (20% exceeding its own gravity), and a large phase difference (around $\gamma = 180°$) which can maintain the hover state with low power consumption (provide lift force approximately equal to gravity). In order to explore the influence of phase difference on hovering, this section investigates the aerodynamic influence of phase difference on TW hovering through the comparative analyses of flow fields under $\gamma = 22.5°$, 135° and 180°, corresponding to the TW hovering with the maximum efficiency and lift ($\gamma = 22.5°$), the minimum efficiency and lift ($\gamma = 135°$) and the most commonly applied condition in observation ($\gamma = 180°$).

Figure 10 shows the $C_L$ and $C_P$ of TF, TH and TW over time at $\gamma = 22.5°$, 135° and 180°, respectively. During the downstroke of TF, the maximum lift is generated at $\gamma = 22.5°$ and the lift force produced by $\gamma = 180°$ is slightly greater than that of 135°. The lift curves of the three-phase differences are almost the same in the upstroke of TF, indicating that the influence of phase differences on the lift of TF is mainly in the downstroke process. Owing to the influence of phase difference, the kinematics of TH at $\gamma = 22.5°$, 135° and 180° are different. Therefore, the lift peak of TH is generated at different time points. In general, under all phase differences, the lift of TH is mainly generated during the

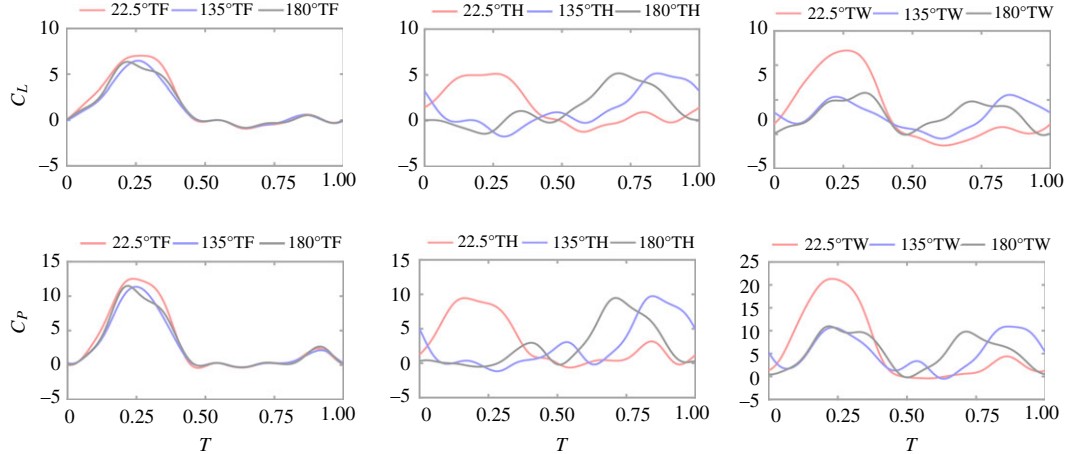

**Figure 10.** $C_L$ and $C_P$ of TF, TH, and TW over time at $\gamma = 22.5°$, 135° and 180°.

**Table 2.** The aerodynamic parameters of tandem-hovering with $\gamma = 22.5°$, 135° and 180°.

|  |  | TF | TH | TW |
|---|---|---|---|---|
| $\gamma = 22.5°$ | CL | 1.78 | 1.60 | 1.69 |
|  | $\eta$ | 54.42 | 53.25 | 53.86 |
| $\gamma = 135°$ | CL | 1.41 (↓20.79%) | 1.29 (↓19.38%) | 1.35 (↓20.11%) |
|  | $\eta$ | 50.83 (↓6.60%) | 49.78 (↓6.52%) | 50.32 (↓6.57%) |
| $\gamma = 180°$ | CL | 1.51 (↓15.17%) | 1.30 (↓18.75%) | 1.41 (↓16.57%) |
|  | $\eta$ | 52.10 (↓4.26%) | 52.44 (↓1.52%) | 52.26 (↓2.97%) |

downstroke, while lift is almost zero during the upstroke. The lift peaks of TH under three-phase differences are almost the same, but hovering with $\gamma = 22.5°$ maintains a large lift near the peak for a long time, making it produce the maximum lift. The lift of TW is the superposition of the lift of TF and TH. Because the TF and TH move almost synchronously under $\gamma = 22.5°$, the lift peak of TW is greater than that of $\gamma = 135°$ and 180°.

The variation law of the $C_P$ is similar to that of the $C_L$, indicating that most of the power is used to generate lift in hover. The time-averaged $C_L$ and $\eta$ of TF, TH and TW at $\gamma = 22.5°$ are all greater than that of $\gamma = 135°$ and 180°. The percentage decrease of the parameters compared with the corresponding data at $\gamma = 25°$ are shown in table 2.

In order to explore the reasons for the great difference in hover aerodynamic performance under different phase differences, this section analyses the vorticity contours and streamlines at $r_2$ section under three-phase differences at $T = 0.3$ and 0.8. As shown in figure 11, when hovering with a small phase difference ($\gamma = 22.5°$), TF and TH are both in the downbeat process ($T = 0.3$), keeping the relative distance unchanged. Therefore, the TEV of TF and LEV of TH maintain a relatively stable structure and strength, which can provide continuous lift. TH is located below TF, which makes the wake of TF less disturbing to the airflow of TH. However, when hovering with $\gamma = 135°$ and 180°, the TEV of TF is affected by the movement of TH, which weakens the strength of the TEV and causes the airflow disturbance of TH. Thus, hovering with $\gamma = 22.5°$ produces greater lift and efficiency than that of $\gamma = 135°$ and 180°.

When hovering with $\gamma = 22.5°$, TF and TH are in upstroke together, parallel to each other at $T = 0.8$, making the wake has less interference with the airflow on the wing surface. Therefore, the vortex structures and streamlines of TF and TH are stable during upstroke. When $\gamma = 135°$ and 180°, the relative movement of TF and TH is obvious, meaning the wake of TF has a great influence on the angle of attack of the incoming flow of TH, which reduces the lift generated by TH. Part of the vortex structure generated by TF during upstroke flows into the LEV of TH, making the LEV structure unstable and causing airflow dissipation, thus reducing the efficiency.

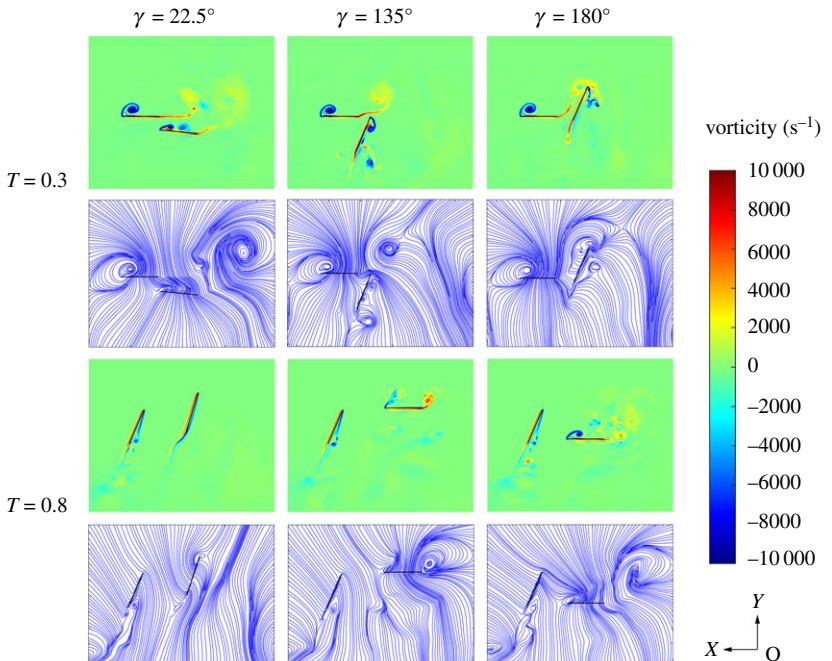

**Figure 11.** Flow fields of TW at $\gamma = 22.5°$, 135° and 180°.

Therefore, it can be seen that in the hovering state, owing to the different phase difference, the TEV of TF and the LEV of TH have different effects on each other. When the phase difference is small, TF and TH beat down and up together, which can not only maintain the stability of vortex structure and intensity, but also help to maintain the high lift force and airflow stability. As the phase difference increases, the relative motion of TF and TH increases, which makes the strength and stability of the LEV of TF and the TEV of TH decrease. So, the lift force and efficiency of hover with $\gamma = 135°$ and 180° decrease compared with that of $\gamma = 22.5°$.

## 5. Conclusion

In this paper, numerical simulation is carried out for TW hovering with $\gamma = 180°$ and the hover of single wing with the same kinematics. Through the comparison of aerodynamic parameters and flow fields, it is shown that when $\gamma = 180°$, the interactions between TF and TH make the lift force and efficiency less than that of the single-wing hovering. The reason is that TF and TH beat in opposite directions at $\gamma = 180°$. When they are close to each other, the vortex structure of TF is partially adsorbed on the TH surface, reducing the strength of vortex ring and aerodynamic force. At the same time, the vortex structure of TF adsorbed by TH is harmful to the stability of the vortex structure, which causes turbulence of airflow, reducing aerodynamic force and efficiency.

The TW hovering with phase difference from 0° to 180° is numerically simulated. By analysing the lift, power and efficiency of different phase differences, the phenomena of different phase difference in dragonfly observation is explained. With phase difference from 0° to 40°, it can provide lift force greater than 20% of the body weight with high efficiency, which is suitable for hovering with payload; at $\gamma = 180°$, the lift supporting the body weight can be provided with low power, which is suitable for long-endurance hovering. The influence of phase difference on the aerodynamic performance of hover is obtained through the flow field of tandem-hovering with $\gamma = 22.5°$, 135° and 180°. The analyses show that under small phase difference, the vortex structure with high stability and strength between the wings is more conducive to the production of high-efficiency and long-endurance lift.

It should be pointed out that the conclusion of this paper is suitable for the hovering state. Whether this conclusion is applicable to other states such as climbing, forward flying, etc. need further study.

Data accessibility. Data available from the Dryad Digital Repository: https://doi.org/10.5061/dryad.prr4xgxkn [43].

Authors' contributions. L.P. designed the study, carried out the numerical simulation, analysed the hovering model and drafted the manuscript. M.Z. carried out wing model reconstruction, participated in data analysis and helped draft the manuscript. T.P. coordinated the study, contributed to data interpretation and helped to conceive of the study.

G.S. participated in designing the study, helped in coordination of the study, helped draft the manuscript and analysis data. Q.L. participated in the design of the study and helped in drafting the manuscript. All authors gave final approval for publication.

Competing interests. We declare we have no competing interests.

Funding. This research is supported by National Natural Science Foundation of China (grant nos. 51636001, 51976005 and 52006002) and Beijing Natural Science Foundation (grant no. 3214047).

Acknowledgements. The authors thank Lab 407 in Beihang University for providing computing resources. The authors thank Xinru Li in LSE for her help in figure processing.

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
