## [Peer Review File · Royal Society Open Science]

Review History

RSOS-202275.R0 (Original submission)

Review form: Reviewer 1

Is the manuscript scientifically sound in its present form?

Yes

Are the interpretations and conclusions justified by the results?

Yes

Is the language acceptable?

Yes

Do you have any ethical concerns with this paper?

No

Have you any concerns about statistical analyses in this paper?

No

Recommendation?

Accept with minor revision (please list in comments)

Comments to the Author(s)

This is an excellent paper and I recommend it for publication with only some very minor suggestions. I found the paper to be very complete. The results are validated with data from well-known and respected published articles. The authors do a great job in explaining their results. I particularly liked the explanation (backed up by the model) of why others have observed two ranges of phase differences for hovering flight in Section 4.3. The figures and tables are excellent and clearly show the results.

Given the authors good ability to explain observed phenomena, I have a few suggestions on other unanswered questions that came up in this paper.

- 1) In the Introduction (Line 44), reference [8] is cited that states dragonflies have the ability to produce an aerodynamic force that 4.3 times greater than their body weight. Yet your analysis showed this to be 1.2 times (20% more). Do you disagree with this reference? How can you explain this discrepancy?
- 2) In Section 3.4, Line 17 you mention total drag as (T_{total}), but elsewhere this term is called thrust. Please be consistent. I assume you meant this to be thrust.
- 3) In Section 4.1 (Line 49) you observe from Figure 4 that lift is primarily generated in the middle portion of the downstroke. It would be good to add a sentence to explain the physicality of why this is an expected result.
- 4) Although the author's do a good job in describing key details of their simulation, providing some additional details on their model would make this an even better paper. In its present form, it would be difficult for a reader to reproduce the model. Some additional detail in this regard would not make the paper too long.

Decision letter (RSOS-202275.R0)

Dear Dr Pan

On behalf of the Editors, we are pleased to inform you that your Manuscript RSOS-202275 "Tandem-wing interactions on aerodynamic performance inspired by dragonfly hovering" has been accepted for publication in Royal Society Open Science subject to minor revision in accordance with the referees' reports. Please find the referees' comments along with any feedback from the Editors below my signature.

Please submit your revised manuscript and required files (see below) no later than 7 days from today's (ie 23-Jun-2021) date. Note: the ScholarOne system will 'lock' if submission of the revision

is attempted 7 or more days after the deadline. If you do not think you will be able to meet this deadline please contact the editorial office immediately.

on behalf of Professor Brooke Flammang (Associate Editor) and Kevin Padian (Subject Editor)
openscience@royalsociety.org

Reviewer comments to Author:
Reviewer: 1

Comments to the Author(s)

This is an excellent paper and I recommend it for publication with only some very minor suggestions. I found the paper to be very complete. The results are validated with data from well-known and respected published articles. The authors do a great job in explaining their results. I particularly liked the explanation (backed up by the model) of why others have observed two ranges of phase differences for hovering flight in Section 4.3. The figures and tables are excellent and clearly show the results.

Given the authors good ability to explain observed phenomena, I have a few suggestions on other unanswered questions that came up in this paper.

- 1) In the Introduction (Line 44), reference [8] is cited that states dragonflies have the ability to produce an aerodynamic force that 4.3 times greater than their body weight. Yet your analysis showed this to be 1.2 times (20% more). Do you disagree with this reference? How can you explain this discrepancy?
- 2) In Section 3.4, Line 17 you mention total drag as (T_{total}), but elsewhere this term is called thrust. Please be consistent. I assume you meant this to be thrust.
- 3) In Section 4.1 (Line 49) you observe from Figure 4 that lift is primarily generated in the middle portion of the downstroke. It would be good to add a sentence to explain the physicality of why this is an expected result.
- 4) Although the author's do a good job in describing key details of their simulation, providing some additional details on their model would make this an even better paper. In its present form, it would be difficult for a reader to reproduce the model. Some additional detail in this regard would not make the paper too long.

===PREPARING YOUR MANUSCRIPT===

===PREPARING YOUR REVISION IN SCHOLARONE===

-- If you have uploaded ESM files, please ensure you follow the guidance at <https://royalsociety.org/journals/authors/author-guidelines/#supplementary-material> to include a suitable title and informative caption. An example of appropriate titling and captioning may be found at https://figshare.com/articles/Table_S2_from_Is_there_a_trade-off_between_peak_performance_and_performance_breadth_across_temperatures_for_aerobic_scops_in_teleost_fishes_/3843624.

Author's Response to Decision Letter for (RSOS-202275.R0)

See Appendix A.

Decision letter (RSOS-202275.R1)

Dear Dr Pan,

I am pleased to inform you that your manuscript entitled "Tandem-wing interactions on aerodynamic performance inspired by dragonfly hovering" is now accepted for publication in Royal Society Open Science.

on behalf of Professor Brooke Flammang (Associate Editor) and Kevin Padian (Subject Editor)
openscience@royalsociety.org

Appendix A

1) In the Introduction (Line 44), reference [8] is cited that states dragonflies have the ability to produce an aerodynamic force that 4.3 times greater than their body weight. Yet your analysis showed this to be 1.2 times (20% more). Do you disagree with this reference? How can you explain this discrepancy?

Response:

I agree with reference. The result of reference is quoted in the Introduction in order to highlight the capacity of dragonflies to generate great aerodynamic force. However, the kinematic state of the dragonfly in reference is the load-lifting process with a small globule of melted soldering tin glued onto the ventral segment, while the dragonfly in this paper is in the hovering state. The kinematic rules of wings are different in the two cases (for example, the rotational angle in this paper is from 65° to 170° , while that in the reference is from 60° to 150°). In my understanding, the dragonfly will actively adjust the kinematic rules of wings when responding to different external conditions, thus generating different aerodynamic forces. Secondly, the aerodynamic force of 4.3 times greater than the body weight in reference is the maximum instantaneous aerodynamic force, and the aerodynamic force of 1.2 times of the body weight in this paper is the time-averaged aerodynamic force. This can also lead to the discrepancy.

2) In Section 3.4, Line 17 you mention total drag as (T_{total}), but elsewhere this term is called thrust. Please be consistent. I assume you meant this to be thrust.

Response:

I agree with you and have changed 'drag' to 'thrust'. In addition, the same mistake also occurred in the other two places, and I have corrected it.

3) In Section 4.1 (Line 49) you observe from Figure 4 that lift is primarily generated in the middle portion of the downstroke. It would be good to add a sentence to explain the physicality of why this is an expected result.

Response:

I think the reason is that dragonflies hover along an inclined stroke plane and the downstroke has a small rotational angle, which makes the aerodynamic force generated during the downstroke is oriented upward and therefore makes a large contribution to the lift. I have added the above explanation to Section 4.1.

4) Although the author's do a good job in describing key details of their simulation, providing some additional details on their model would make this an even better paper. In its present form, it would be difficult for a reader to reproduce the model. Some additional detail in this regard would not make the paper too long.

Response:

In order to make it easier for readers to reproduce the model, the geometry files used in the model are uploaded in the electronic supplementary material (ESM). At the same time, the specific setting of the model, such as the wall model, the refinement method, the size of mesh near the boundary and the way to define kinematic rules of the wings are explained in the corresponding positions of Section 3.2 and Section 3.3.